# The Relationship between Parenting Behaviors and Adolescent Well-Being Varies with the Consistency of Parent–Adolescent Cultural Orientation

**DOI:** 10.3390/bs14030193

**Published:** 2024-02-28

**Authors:** Tixiang Yang, Xiaosong Gai, Su Wang, Stanley Gai

**Affiliations:** 1School of Psychology, Northeast Normal University, 5268 Renmin Street, Changchun 130024, China; yangtx996@nenu.edu.cn (T.Y.); wangs427@nenu.edu.cn (S.W.); 2Research Center of Mental Health Education in Northeast Normal University, Key Research Institute of Humanities and Social Science in Universities in Jilin Province, Changchun 130024, China; 3Freeman College of Management, Bucknell University, Lewisburg, PA 17837, USA; stanley.gai@bucknell.edu

**Keywords:** cultural orientation, parenting behaviors, adolescent basic psychological needs, adolescent well-being

## Abstract

To evaluate the limitations of the traditional parenting model in the cultural transition period, this study investigated the relationship between parenting behaviors and adolescents’ well-being, in which the moderating role of consistency in parent–adolescent cultural orientation was also investigated. Six hundred forty-four parent–adolescent dyads completed self-report surveys. Parents completed the cultural orientation questionnaire (parental version), and adolescents completed the cultural orientation questionnaire (adolescent version), the adolescent-perceived parenting behavior scale, the adolescent basic psychological needs scale, and the adolescent well-being questionnaire. The findings were as follows: (1) Adolescent-perceived parental autonomy support positively predicted the satisfaction of adolescents’ basic psychological needs, thereby enhancing adolescents’ well-being levels. Conversely, adolescent-perceived parental control significantly predicted the frustration of adolescents’ basic psychological needs, thereby reducing their well-being levels. (2) When both parents and adolescents share a collectivistic cultural orientation, high parental control significantly frustrated adolescents’ basic psychological needs, but it did not negatively affect their well-being. However, when parents are collectivists but adolescents are individualists, high parental control would significantly induce the frustration of basic psychological needs, thus further impairing adolescents’ well-being. The results revealed that differences in cultural orientations between generations during cultural transition periods moderate the effects of parenting behaviors.

## 1. Introduction

Parenting behaviors significantly impact the satisfaction of adolescents’ basic psychological needs, consequently affecting their developmental outcomes. Based on self-determination theory (SDT) [1], autonomy-supportive parenting satisfies children’s and adolescents’ basic psychological needs, while high-control parenting leads to their needs being frustrated. Autonomy support from parents is positively associated with the satisfaction of basic psychological needs in children and adolescents aged 9 to 20 [2], and the satisfaction of these needs further enhances children’s and adolescents’ well-being [1,3,4,5,6]. The satisfaction of adolescents’ basic psychological needs positively influences their life satisfaction [7], vitality [8], and daily well-being [9]. Conversely, there is a positive correlation between parental control and the occurrence of psychological illnesses and problem behaviors in adolescents [10,11,12]. The frustration of adolescents’ basic psychological needs will further lead to maladjustment and psychological illnesses [13,14,15]. Furthermore, the frustration of these needs negatively impacts adolescents’ life satisfaction, depression [16,17], and anxiety [18].

Parental cultural orientation has an impact on parenting behaviors. Chinese traditional culture is viewed as collectivist, while Western cultures tend to reflect individualism. Consequently, Western parents are more likely to foster independence and individualized characters in their children, emphasizing the development of a self that is distinct from the family during the parenting process. In contrast, Chinese parents are likely to promote a self that is interdependent with the family, community, and society, focusing on the collective characteristics of their children [19]. Mothers with an independent cultural orientation are more likely to report goals of promoting children’s self-development and independence and are more likely to identify with autonomy-supportive parenting behaviors. Conversely, mothers with an interdependent cultural orientation are more likely to report goals and behaviors related to fostering children’s relational development with others [20]. Parents from individualistic cultures tend to listen to their children’s opinions, respect their ideas, and encourage free choice. In contrast, those from collectivistic cultures often require children to respect and obey authority figures [21]. Therefore, compared to parents from individualistic cultural backgrounds, parents from collectivistic cultural backgrounds are more likely to exhibit high-control parenting behaviors [22,23,24,25].

Adolescents’ cultural orientation moderates their developmental outcomes and the effects of parenting behaviors. King et al. (2015) conducted a study on 466 Filipino university students to investigate the direct effects of family obligations on students’ learning motivation (autonomous vs. controlled), school engagement (behavioral and emotional), and well-being (life satisfaction and positive and negative emotions). Additionally, they examined the moderating role of the students’ interdependent self-construal. They found that compared to students with independent self-construal, those with interdependent self-construal experienced more positive effects of family obligations on autonomous learning motivation and life satisfaction. Moreover, family obligations had no significant effect on school engagement and the well-being of students with independent self-construal. However, family obligations negatively predicted lower school engagement and well-being levels in students with interdependent self-construal. This suggests that for students with interdependent self-construal, family obligations can significantly enhance their autonomous learning motivation, well-being, and school engagement [26]. Pierre et al. (2019) conducted a study on 245 American and 156 Ghanaian adolescents to examine the impact of perceived parental autonomy support on adolescents’ development (school engagement, learning motivation, self-worth and depression) and the moderating role of adolescents’ self-construal (independent–interdependent). The findings indicated that, compared to adolescents with interdependent self-construal, those with independent self-construal perceived a stronger negative relationship between parental control and parental autonomy support. Additionally, perceived parental autonomy support positively impacted school engagement for adolescents with independent self-construal. In the relationship between perceived parental autonomy support and adolescents’ depression, perceived autonomy support was positively related to depression in adolescents with interdependent self-construal but unrelated to those with independent self-construal [27].

During rapid social change, there are noticeable differences in cultural orientation between generations. Younger generations tend to display more individualistic tendencies than older ones [28]. Ren and Xu (2015) used objective indicators, such as divorce rates and urbanization levels, as well as subjective indicators, like social trust and respect for parents, as measures of individualism and GDP as a modernization indicator. They analyzed the trend of individualism in Chinese people from 1980 to 2010 using official data, finding a generational difference in individualism among Chinese people, indicating a gradually increasing trend over the 30 years [29]. Ma et al. (2016) surveyed the values of workers of different ages and found that the younger generation (particularly those born after the 1990s) scored higher on individualism and lower on collectivism [30]. Shen et al. (2017) used the “Rokeach Value Survey” in 2015 to conduct a value survey on 1464 adolescents in six cities in China, including Beijing, Guangzhou, and Jilin. They compared the results with those from 1987, 1998, and 2004. They found that the ultimate values of adolescents have further strengthened in terms of individualism and pragmatism. Overall, these studies highlight a trend toward increasing individualism and decreasing collectivism among younger generations in China, particularly during periods of rapid social change [31].

### This Study

Cultural orientation influences parents’ parenting behavior. Parents under the influence of a collectivist cultural background emphasize concepts such as hierarchy, filial piety, parental authority, and self-restraint [22,25]. Therefore, we speculate that Chinese parents will participate too much in the parenting process and use psychological control to ensure that their children’s growth meets social requirements and severely punish behaviors that violate social norms. However, in the study of parental autonomy support and parental control on adolescent well-being, only a single measurement of parental control parenting behavior is not comprehensive. Therefore, combined with the social and cultural background of Chinese collectivism, this study took psychological control, parental over-protection, and severe punishment as comprehensive indicators of parental control. According to self-determination theory, parental autonomy support will have a positive impact on the satisfaction of adolescents’ basic psychological needs, and parental control of parenting will have a positive impact on the frustration of adolescents’ basic psychological needs [1]. As a core element of well-being, basic psychological needs can predict individual well-being regardless of cultural background [5,32]. Therefore, this study proposes the first hypothesis that parental autonomy support can predict the satisfaction of adolescents’ basic psychological needs and thus enhance their well-being. Parental control will lead to the frustration of adolescents’ basic psychological needs and reduce the level of adolescents’ well-being.

In the past few decades, China has experienced rapid economic development and social transformation, and the change in cultural orientation is one of the manifestations of rapid social change. With the changes in society, individualism has gradually increased in China, while collectivism has gradually declined [33]. Chinese parents also began to find a balance between traditional cultural values and individualistic values under the influence of globalization [34]. Some Chinese parents may cling to a collectivist cultural orientation, while other Chinese parents may be more inclined to an individualist cultural orientation in the process of social change. Moreover, the cultural orientation held by adolescents growing up under evolving social institutions and peer socialization is also dynamically developing, and it has gradually changed from a collectivist cultural orientation to an individualist cultural orientation [31]. This will ultimately impact the parent–adolescent interactions since their cultural orientation can either be consistent or inconsistent. In parenting research, it has been proved that parents in a collectivist society will have a higher level of control, while parents in an individualistic culture will have a higher level of autonomy [22,35]. Whether this conclusion can be inferred at the individual level deserves further investigation. Assuming that this conclusion can be inferred at the individual level, we further make a guess about whether the parental control implemented by collectivist parents and the parental control implemented by individualistic parents have different effects on the well-being of adolescents with different cultural orientations. A few studies have confirmed the moderating effect of adolescents’ cultural orientation on adolescent-perceived parenting behavior and its developmental results [27], while the moderating effect of parent–adolescent cultural orientation on adolescent-perceived parenting behaviors and its developmental results has not been fully confirmed. Therefore, this study aims to further investigate the significance of consistency or discrepancy in parent–adolescent cultural orientation on parenting behaviors and adolescents’ well-being. We propose the following second hypothesis: the relation between adolescent-perceived parental control and their well-being will be moderated by the consistency or discrepancy of parent–adolescent cultural orientation. Specifically, compared with parents who are collectivists but adolescents who are individualists, when parents and adolescents are both collectivistic cultural orientations, the frustration of basic psychological needs caused by adolescents’ perception of parental control has less impact on adolescents’ well-being.

## 2. Materials and Methods

### 2.1. Participants

This study conducted a survey using a random sampling method to collect data from students and their parents in six schools across Jilin Province, Sichuan Province, and Guangdong Province. The survey yielded a dataset of 644 valid parent–adolescent pairs, with 348 female students (54%) and 296 male students (46%).

The survey included 493 mothers (63%), 213 fathers (33%), and 28 other caregivers (4%). The educational distribution of the surveyed parents was as follows: 71 with primary education or less (11%), 256 with junior high school education (40%), 116 with high school or technical secondary school education (18%), 82 with junior college education (13%), and 119 with bachelor’s degree or higher (18%). The occupational distribution of parents was as follows: 92 were unemployed or semi-unemployed (14%), 265 were in service industries or manual labor (41%), 37 were clerical workers (6%), 106 were self-employed with no or few employees (17%), 16 were owners of medium or large businesses (3%), 38 held middle management positions in enterprises (6%), 77 were professionals (12%), and 13 were government employees (2%). In summary, this study’s results tend to reflect the cognition of mothers from the middle and lower social strata.

### 2.2. Measures

#### 2.2.1. Parental Cultural Orientation Questionnaire

In order to make the content of the questionnaire in line with Chinese cultural background and family characteristics, this study adapted the Chinese Parents’ Cultural Orientation Questionnaire on the basis of sixteen cultural orientation measurement tools. Based on the two-level dimensions of individualism–collectivism proposed by Oyserman et al.’s meta-analysis [36] and the method of Hui et al. to examine individual cultural orientation by dividing groups [37], we adopted the forced election form used in Fisher et al.’s study [38], supplemented by the relevant dimensions and questions in the Asian Values Scale developed and improved in the study by Kim [39,40]. The adaptation of the parental cultural orientation questionnaire was finally completed.

The “Parental Cultural Orientation Questionnaire” comprises 40 items across six dimensions: personal development (4 items), family relations (9 items), filial piety concepts (5 items), emotional communication (5 items), privacy protection (3 items), and parental role (14 items) (Questionnaire detailed in Appendix A). It utilizes a forced-choice format, where selecting an individualistic cultural orientation option scores positive one point, selecting a collectivistic cultural orientation option scores negative one point, and selecting an uncertain option scores 0 points. The total score from all 40 items is then summed and averaged. A negative average indicates a tendency towards collectivism, while a positive average indicates a tendency towards individualism within the familial domain. The greater the absolute value of the final score, the stronger the tendency towards that cultural orientation. In this study, the internal consistency coefficient of the “Parental Cultural Orientation Questionnaire” is 0.732. The questionnaire’s discrimination index was below 0.2 for 1 item (3%), between 0.2 and 0.4 for 12 items (30%), and above 0.4 for 27 items (67%).

#### 2.2.2. Adolescents’ Cultural Orientation Questionnaire

The “Adolescents’ Cultural Orientation Questionnaire” was adapted from the “Parental Cultural Orientation Questionnaire” and includes 30 items across seven dimensions: personal development (3 items), family relations (6 items), filial piety concepts (4 items), emotional communication (5 items), privacy protection (3 items), parental role (6 items), and pursuit of individuality (3 items). The scoring method is the same as the Parental Cultural Orientation Questionnaire. In this study, the consistency coefficient for the Adolescents’ Cultural Orientation Questionnaire is 0.624. The discrimination index of the questionnaire was below 0.2 for 1 item, accounting for 3%; between 0.3 and 0.4 for 3 items, accounting for 10%; and 0.4 or above for 26 items, accounting for 87%.

#### 2.2.3. Adolescents’ Perception of Parenting Behaviors Questionnaire

Based on the two-factor autonomy support scale of Wang et al. (2007) [41] and the three-factor autonomy support scale of Mageau et al. (2015) [36], as well as other tools like the Perceived Parental Autonomy Support Scale [42] and the Chinese version of Parenting Styles and Dimensions Questionnaire (PSDQ) [43], relevant items for the parental autonomy support scale were adapted. Additionally, based on three-factor psychological control scale (guilt induction, love withdrawal, and authoritarianism) of Wang et al. (2007) [41] and five-factor psychological control scale (physical attack, love withdrawal, over-control, verbal inhibition and invalidating feelings) of Shek (2005, 2008) [44,45], combined with tools like the Children’s Report of Parental Behavior Inventory (CRPBI) [46] and Psychological Control Scale -Youth Self-Report [47], relevant items for the psychological control subscale were adapted. It also absorbed and adapted from the Overparenting Scale [48,49,50,51] and the Authoritarian Subscale [43] to form the Adolescents’ Perception of Parenting Behaviors Questionnaire.

The questionnaire comprises two subscales: autonomy support and parental control. The autonomy support subscale includes encouragement of autonomy (7 items), acknowledgment of points and feelings (3 items), and rational induction (3 items), totaling 13 items. The parental control subscale includes psychological control (9 items), over-control (3 items), and coercion (8 items), totaling 20 items, making 33 items in total. It uses a 5-point rating scale, and the average scores of the autonomy support and parental control subscales are calculated separately; higher scores indicate higher levels of perceived parental autonomy support or control. In this study, the internal consistency coefficient for the Adolescents’ Perception of Parental Autonomy Support Scale is 0.924, and for the Adolescents’ Perception of Parental Control Scale, it is also 0.924. The structural validity of the perceived parental autonomy support scale is indicated by x^2^/*df* = 1.463, CFI = 0.995, TLI = 0.992 and RMSEA = 0.027; and for the perceived parental control scale it is indicated by x^2^/*df* = 1.558, CFI = 0.985, TLI = 0.980 and RMSEA = 0.029.

#### 2.2.4. Basic Psychological Needs Questionnaire

The questionnaire was revised by Chen et al. (2015) [16], encompassing three dimensions: autonomy, relatedness, and competence needs. Each dimension contains 8 items, assessing the satisfaction and frustration of these needs. Specifically, the satisfaction and frustration of autonomy, relatedness, and competence needs are each assessed by using 4 items. A 5-point scoring system is used, and the score for each dimension is the average of its corresponding items. A higher average score indicates a higher level of either satisfaction or frustration for the respective dimension. In this study, the internal consistency coefficient for the Basic Psychological Needs Satisfaction Scale is 0.824, and for the Basic Psychological Needs Frustration Scale is 0.858.

#### 2.2.5. Adolescents’ Well-Being Questionnaire

The questionnaire utilizes the Adolescents’ Well-Being Questionnaire compiled by Wang Wen et al. (2015; 2018) [52,53]. The questionnaire was refined based on factor loadings without altering the original dimensions or items. It still comprises six dimensions: current life satisfaction (9 items), current positive affect (3 items), current negative affect (3 items), future expectation satisfaction (5 items), future positive affect (3 items), and future negative affect (3 items), totaling 26 items. A 5-point rating scale is used, and the score for each dimension is the average of its corresponding items. Higher average scores indicate higher levels of well-being in the respective dimension. In this study, the internal consistency coefficient for the Adolescents’ Well-Being Questionnaire is 0.819. The structural validity of the questionnaire was tested among participants using Amos 26.0 and the maximum-likelihood (ML) method. The revised model’s fit indices are x^2^/*df* = 1.675, CFI = 0.978, TLI = 0.972 and RMSEA = 0.032.

The overall well-being score for adolescents is synthesized following the method used in the study by Wu Xiaojing (2021), which is expressed as follows: Overall Well-Being = [Current Satisfaction + (Current Positive Affect − Current Negative Affect + 6) ∗ 0.5 + Future Expectation Satisfaction + (Future Positive Affect − Future Negative Affect + 6) ∗ 0.5]/4 [49]. This formula balances current and future aspects of well-being, incorporating both affective and cognitive dimensions [54].

### 2.3. Procedure

From the end of June to the beginning of July 2023, paper questionnaires were packaged and sent to psychological teachers at various schools. The teachers distributed two copies of the questionnaire to students on a Friday (one for the student to complete and one for the parent). Students were instructed to take the questionnaires home for both themselves and their parents to complete. The following four points were emphasized to the students:(1)Both the student and parent versions of the questionnaire should include the student’s school ID or the last four digits of their National Identification Number to ensure that each student’s and parent’s responses can be matched accurately.(2)The student version of the questionnaire must be completed by the student themselves and not by someone else. The parental version should be completed by a parent. If the parent is away for work, another guardian at home may answer it, but under no circumstances should the student fill it out on behalf of the parent.(3)There are no right or wrong answers to any of the questions; participants should answer according to their actual situations.(4)All provided information will be kept strictly confidential.

This procedure ensures that the data collected reflects genuine and independent responses from both the adolescents and their parents, which is crucial for the validity and reliability of this study’s findings.

### 2.4. Data Analyses

Before analyzing the data, we cleaned the data according to the following criteria:(1)Delete the questionnaires whose informed consent right is ‘No’;(2)Delete the questionnaires with missing or wrong answers;(3)Remove the questionnaires with a total score of 4 points or more of lie detection questions in the Parental Cultural Orientation Questionnaire;(4)Questionnaires with a total lie detection score of 4 or above were deleted from the questionnaires related to adolescents’ cultural orientation, perceived parenting behavior, basic psychological needs satisfaction, and well-being;(5)Either the student ID number or student number and the class filled in the Parental Cultural Orientation Questionnaire and the Adolescent Cultural Orientation Questionnaire can have a one-to-one correspondence. If not, parents and the youth version of the cultural orientation questionnaire will be deleted.

After data cleaning, there were no missing values in the dataset. Based on this, data was analyzed using SPSS 23.0 for descriptive statistics, correlation analysis, and analysis of variance. Amos 26.0 was employed for validity testing, as well as mediation and moderation effect testing.

## 3. Results

### 3.1. Test for Common Method Bias

An exploratory factor analysis was conducted on all variables in the questionnaires. The results revealed that, without rotation, there were 27 factors with eigenvalues bigger than 1. The first factor explained 19.26% of the variance, which is less than the 40% threshold often used to indicate significant common method bias. This suggests that the level of common method bias in this study is acceptable and does not significantly distort the results [55].

### 3.2. Descriptive Statistics and Correlation Analysis

A Pearson correlation analysis was conducted on these variables to investigate the relationships among parental and adolescents’ cultural orientations, adolescents’ perceptions of parental autonomy support and control, the satisfaction and frustration of adolescents’ basic psychological needs, and adolescents’ well-being. The relationships between each of the variables are presented in Table 1. The results indicated the following:(1)Parental cultural orientation (individualism–collectivism) is positively correlated with adolescents’ cultural orientation (individualism–collectivism).(2)Parental cultural orientation (individualism–collectivism) is positively correlated with adolescent-perceived parental autonomy support, adolescent basic psychological needs, and adolescent well-being. It is negatively correlated with adolescent-perceived parental control and the frustration of adolescents’ basic psychological needs.(3)Adolescents’ cultural orientation (individualism–collectivism) is positively correlated with adolescent basic psychological needs satisfaction and adolescents’ well-being but negatively correlated with the frustration of adolescent basic psychological needs.(4)Adolescent-perceived parental autonomy support is positively correlated with adolescent basic psychological needs satisfaction and adolescents’ well-being but negatively correlated with adolescent-perceived parental control and the frustration of adolescent basic psychological needs.(5)Adolescent-perceived parental control is positively correlated with the frustration of adolescent basic psychological needs but negatively correlated with adolescent basic psychological needs satisfaction and adolescents’ well-being.(6)Adolescent basic psychological needs satisfaction is positively correlated with adolescents’ well-being but negatively correlated with the frustration of basic psychological needs.

**Table 1 behavsci-14-00193-t001:** The mean, standard deviation, and correlation coefficient of each variable (*n* = 644).

Variable	M	SD	1	2	3	4	5	6	7
1. PCO(I-C)	0.18	0.24							
2. ACO(I-C)	0.13	0.22	0.242 **						
3. APAS	3.84	0.87	0.215 **	0.070					
4. APPC	2.70	0.86	−0.233 **	−0.127 **	−0.636 **				
5. ABPNS	3.69	0.68	0.108 **	0.202 **	0.472 **	−0.350 **			
6. ABPNF	2.83	0.82	−0.212 **	−0.175 **	−0.391 **	0.574 **	−0.435 **		
7. AWB	3.51	0.74	0.109 **	0.142 **	0.509 **	−0.419 **	0.646 **	−0.586 **	

Note: ** *p* < 0.01. PCO(I-C) = parental cultural orientation (individualism–collectivism); ACO(I-C) = adolescents’ cultural orientation (individualism–collectivism); APAS = adolescent-perceived autonomy support; APPC = adolescent-perceived parental control; ABPNS = adolescent basic psychological needs satisfaction; ABPNF = adolescent basic psychological needs frustration; AWB = adolescents’ well-being.

The research results imply that the cultural orientations of parents and adolescents and parenting behaviors impacted adolescents’ general well-being. This suggests that the data is appropriate for further analysis, revealing significant correlations between cultural orientations, perceived parenting behaviors, basic psychological needs, and adolescents’ well-being.

### 3.3. The Impact of Parenting Behavior on Adolescents’ Basic Psychological Needs and Well-Being

The results indicated significant correlations between parental cultural orientation, adolescent-perceived autonomy support and parental control, adolescent basic psychological needs satisfaction and frustration, and adolescents’ well-being. Based on these correlations, it is deduced that adolescent basic psychological needs warrant further analysis as potential mediation.

Due to limitations such as lack of error analysis and the fixed number of independent and dependent variables in the SPSS built-in plugin PROCESS, this study opted to use Amos 26.0 to construct a structural equation model. This model is used to test the mediating role of basic psychological needs. By utilizing Amos 26.0, this study can more accurately estimate the complex relationships and indirect effects between variables, providing a comprehensive understanding of how parenting behaviors influence adolescents’ overall well-being through basic psychological needs. This approach allows for a nuanced exploration of the pathways through which parenting behaviors impact adolescent outcomes.

When analyzing the mediating pathways of adolescent basic psychological needs in the impact of parenting behaviors on adolescents’ well-being using Amos 26.0, the overall sample’s model fit indices were found to be CFI = 1.0, NFI = 1.0 and GFI = 1.0, indicating an excellent model fit. Further path analysis revealed the following significant relationships: (1) Adolescent-perceived autonomy support was significantly positively related to the satisfaction of adolescent basic psychological needs (β = 0.418, *p* < 0.001). (2) Adolescent-perceived parental control was significantly positively related to the frustration of adolescent basic psychological needs (β = 0.546, *p* < 0.001). (3) The satisfaction of adolescent basic psychological needs was significantly positively related to adolescents’ well-being (β = 0.410, *p* < 0.001). (4) The frustration of adolescents’ basic psychological needs was significantly negatively related to adolescents’ well-being (β = −0.364, *p* < 0.001). (5) Adolescent-perceived parental autonomy support was significantly positively related to adolescents’ well-being (β = 0.223, *p* < 0.001), while parental control was not significantly related to adolescents’ well-being (β = 0.074, *p* > 0.05).

The significance level of the mediating effects of adolescent basic psychological needs between parenting behaviors and adolescents’ well-being was estimated using a non-parametric bootstrap test. This involved 2000 resamples with replacements to create a sampling distribution. The results of the mediation effect analysis (as indicated in Table 2 and Figure 1) show that adolescent basic psychological needs satisfaction mediated the relationship between adolescent-perceived autonomy support and their well-being. Adolescent basic psychological needs frustration mediated the relationship between adolescent-perceived parental control and their well-being. Specifically, the mediating effects comprise two indirect effects pathways: The first indirect effect (0.171) was generated through the pathway involving adolescent-perceived autonomy support → adolescents’ basic psychological satisfaction → adolescents’ well-being. The second indirect effect (−0.199) was generated through the pathway involving adolescent-perceived parental control → adolescents’ basic psychological frustration → adolescents’ well-being.

Bootstrap 95% confidence intervals of both pathways did not include zero, indicating that both indirect effects are statistically significant. This means that both the satisfaction and frustration of adolescents’ basic psychological needs are significant mediators in how parenting behaviors (either supportive or controlling) translate to adolescents’ well-being. These findings reinforce the critical roles of autonomy support and psychological needs satisfaction in fostering youth positive development, as well as the detrimental effects of parental control and psychological needs frustration.

### 3.4. The Moderating Role of the Consistency of Parent–Adolescent Cultural Orientation in the Relationship between Parenting Behaviors and Adolescent Well-Being

Based on the scores of the parental and adolescents’ cultural orientation questionnaires, the cultural orientation of parents and adolescents can be divided into two categories: individualism and collectivism. The cultural orientation of parents and adolescents was coded and paired. Therefore, four types of parent–adolescent cultural orientations were identified: In type I, both the parent and the adolescent have collectivistic cultural orientations. In this alignment, both parent and adolescent share similar collectivistic values, potentially leading to more harmonious interactions and understandings related to collectivistic practices and expectations. In type II, the parent has a collectivistic cultural orientation, while the adolescent has an individualistic cultural orientation. This mismatch might lead to conflicts or misunderstandings, as the parental emphasis on collectivistic values might clash with the adolescent’s individuality and autonomy. In type III, both the parent and the adolescent have individualistic cultural orientations. Here, both share similar values emphasizing personal freedom, independence, and self-expression, which might lead to greater autonomy support and understanding between the two. In type IV, the parent has an individualistic cultural orientation, while the adolescent has a collectivistic cultural orientation. This situation might be rarer but can occur, especially in contexts of cultural change or in multi-cultural families, leading to potential conflicts or a unique blending of values.

Multi-group analysis was conducted to identify whether the path coefficients differ significantly in the consistency or inconsistency in parent–adolescent cultural orientation. An unconstrained model, a structural weight model and a structural residual model were developed, respectively. As displayed in Table 3, the results showed that the constrained models (structural weight model and structural residual model) were significantly different from the unconstrained model (*p* < 0.05), suggesting significant differences among different types of parent–adolescent cultural orientations. Further comparing the difference in the coefficients, we found the consistency or inconsistency of parent–adolescent cultural orientation played a role in the following pathways.

Table 4 presents the standardized path coefficients in different groups, as well as Table 5 shows critical ratios for differences between parameters. From the satisfaction of adolescents’ basic psychological needs to adolescents’ well-being, the path coefficients were 0.276 (*p* < 0.01) and 0.535 (*p* < 0.001) for the type Ⅱ (parents are collectivists but adolescents are individualists) and type Ⅳ (parents are individualists but adolescents are collectivists), respectively. The absolute value of critical ratios for differences between parameters was 2.098 (>1.96), significantly different at the 0.05 level. When parents are individualists but adolescents are collectivists, the positive predictive effect of adolescents’ basic psychological need satisfaction on adolescents’ well-being is stronger than that when parents are collectivists but adolescents are individualists.

From adolescents’ basic psychological need frustration to adolescents’ well-being, the path coefficients were −0.212 (*p* > 0.05) and −0.561 (*p* < 0.01) for the type Ⅰ (parents and adolescents are both collectivists) and type Ⅱ (parents are collectivists but adolescents are individualists), respectively. The absolute value of critical ratios for differences between parameters was 3.154 (>1.96), which was significantly different at the 0.05 level. Compared with parents who are collectivists but adolescents who are individualists, when parents and adolescents are both collectivistic cultural orientations, the negative impact of adolescents’ basic psychological need frustration on adolescents’ well-being is not significant.

From adolescents’ basic psychological need frustration to adolescents’ well-being, the path coefficients were −0.561 (*p* < 0.01) and −0.372 (*p* < 0.001) for the type Ⅱ (parents are collectivists but adolescents are individualists) and type Ⅲ (parents and adolescents are both individualists), respectively. The absolute value of critical ratios for differences between parameters was 2.421 (>1.96), which was significantly different at the 0.05 level. When parents are collectivists but adolescents are individualists, the frustration of adolescents’ basic psychological needs has a greater negative impact on adolescents’ well-being compared to the case in which parents and adolescents are both individualists.

From adolescents’ basic psychological need frustration to adolescents’ well-being, the path coefficients were −0.561 (*p* < 0.01) and −0.345 (*p* < 0.001) for the type Ⅱ (parents are collectivists but adolescents are individualists) and type Ⅳ (parents are individualists but adolescents are collectivists), respectively. The absolute value of critical ratios for differences between parameters was 2.143 (>1.96), which was significantly different at the 0.05 level. When parents are collectivists but adolescents are individualists, the frustration of adolescents’ basic psychological needs has a greater negative impact on adolescents’ well-being compared to the case of parents who are individualists but adolescents are collectivists.

According to Figure 2, Figure 3, Figure 4 and Figure 5, among the four types of parent–adolescent cultural orientation, whether there is consistency in cultural orientation between parents and adolescents or not, parents still recognize and implement autonomy-supportive parenting behaviors. This is because autonomy support positively impacts adolescents’ satisfaction of basic psychological needs and their overall well-being. While high parental control can lead to the frustration of adolescents’ basic psychological needs, it does not necessarily reduce adolescent well-being in the context of a shared collectivist orientation.

However, the relationship between adolescent-perceived parental control and adolescents’ well-being varies as the parent–adolescent cultural orientation. Compared with parents who are collectivists but adolescents who are individualists, when parents and adolescents are both collectivists, even though high-control parenting behavior significantly exacerbates the frustration of adolescents’ basic psychological needs, it does not significantly negatively affect their well-being. When parents are collectivists but adolescents are individualists, high-control parenting behavior will not only significantly aggravate the frustration of basic psychological needs but also further lead to a significant reduction in adolescents’ well-being.

This finding suggests that autonomy support is still valued and beneficial for adolescent psychological outcomes within collectivist cultural contexts. It also implies a potential resilience or mitigating effect within collectivist families, where high control might not have as detrimental an impact on adolescent well-being as it might in families with differing or individualistic cultural orientations. This nuanced view highlights the complexity of parenting strategies and their outcomes within specific cultural frameworks, emphasizing the importance of considering cultural nuances when understanding and advising on parenting approaches.

## 4. Discussion

### 4.1. Reaffirmation of the Role of Basic Psychological Needs in the Relationship between Parenting Behavior and Adolescent Well-Being

Building on self-determination theory [1], this study reconfirms the mediating role of basic psychological needs satisfaction between parental autonomy support and adolescents’ well-being, as well as the mediating role of basic psychological needs frustration between parental control and adolescents’ well-being. This is consistent with the findings of existing research [56,57,58].

Initially, adolescents’ perception of parental autonomy support positively impacts their well-being. This result can be interpreted as follows: (a) parental autonomy support allows adolescents to feel more supported and cared for; (b) by supporting adolescents’ autonomous choices and decisions, parents communicate their trust in adolescents, fostering a sense of competence in adolescents [59]; and (c) parental autonomy support enhances adolescents’ intrinsic motivation [60,61]. That is, when adolescents perceive an external environment supportive of autonomy, they are encouraged by the environment, aiding them in fully utilizing internal resources, proactively adapting to their surroundings, and achieving self-development. In this process, adolescents are likely to experience heightened well-being. Thus, this result suggests that parents should actively provide autonomy support during the parenting process.

Furthermore, adolescents’ perception of parental control leads to frustration with their basic psychological needs and subsequently decreases their well-being. This might be because parents enforce standards without considering the adolescents’ values and needs [62], which can undermine adolescents’ autonomy and hinder their ability to establish and express their thoughts and feelings [63]. This can lead to feelings of inferiority and make adolescents prone to subjective distress and negative emotions [64].

These findings suggest that the satisfaction and frustration of basic psychological needs are crucial mediators in the relationship between parenting behaviors and adolescents’ well-being. Specifically, adolescent-perceived parental autonomy support is beneficial for the satisfaction of psychological needs and subsequently enhances well-being, whereas adolescent-perceived parental control contributes to the frustration of these needs, detracting from well-being. The direct effect of perceived autonomy support on well-being is also significant, highlighting the importance of supportive parenting behaviors in promoting adolescents’ well-being. Conversely, the direct relationship between parental control and adolescent well-being was not significant, suggesting that its influence is primarily exerted through the frustration of basic psychological needs.

### 4.2. Providing Initial Evidence for the Moderating Role of Parent–Child Cultural Orientation Alignment in the Outcomes of Parental Control Behaviors

A common characteristic of Chinese parenting is a high level of parental control [65]. Deeply influenced by the Chinese parenting notion of “guan” or governance, Chinese parents often express their care and support through controlling measures. Due to the influence of this cultural norm, Chinese adolescents might perceive parental control and over-involvement as expressions of love and care [22,66]. However, according to self-determination theory (SDT), parental control characterized by dominance and intrusiveness is related to negative developmental outcomes for adolescents [17,41]. This leads to the following question: is the consequence of parental control on adolescents’ well-being culturally specific? This question has been a long-standing and contentious issue in the field of cross-cultural parenting research.

This study addressed the question of cultural specificity in the impact of parental control on adolescents’ well-being by examining the consistency in parent–adolescent cultural orientation. On the one hand, when parents are collectivists but adolescents are individualists, adolescent-perceived parental control will lead to the frustration of their basic psychological needs and thus reduce their level of well-being. This is consistent with self-determination theory [1]. On the other hand, when parents and adolescents both hold a collectivistic orientation, even if high-control parenting behavior significantly aggravates the frustration of adolescents’ basic psychological needs, it does not significantly negatively affect their well-being. This can be interpreted using the person-environment fit theory introduced by Murray and Lewin in management studies, which emphasizes that individual values and characteristics affect the degree of person-environment fit [67]. When there is a match, it promotes positive attitudes and behaviors in individuals, such as higher job satisfaction, less stress, and better job performance [68]. This suggests that adolescents’ perception and reaction to parental control may depend on whether their cultural orientation aligns with their parents.

Therefore, when parents and adolescents both share a collectivistic cultural orientation, adolescents may view parental control as an expression of care and love [69]; hence, they might have a more positive appraisal of parental control [70], perceive a lower level of parental control [71], and experience less autonomy frustration [72] compared to the insistency of parent–adolescent cultural orientation. Moreover, in response to parental control, adolescents with collectivist cultural orientation might comply with their parents’ controlling behaviors, facilitating harmonious parent–adolescent relationships, which may be beneficial for the satisfaction of adolescents’ related needs.

Conversely, for the situation that parents are collectivistic but adolescents are individualistic cultural orientation, we can make the following explanation: firstly, collectivist parents tend to be highly controlling and pay more attention to supervising adolescents, which will greatly dampen the autonomy of adolescents with individualistic cultural orientation, resulting in their limited choice and decision-making. Secondly, due to cultural clashes with collectivistic parents, individualistic adolescents are more likely to view parental control as an infringement on their autonomy and feel more stress in the family environment, which exacerbates their autonomy frustration and impedes their autonomous development, ultimately intensifying the frustration of basic psychological needs. Thirdly, collectivistic parents affected by traditional culture will have high expectations for adolescents or excessive supervision of adolescents’ behavior, which may make adolescents with individualistic cultural orientation feel frustrated and anxious because they cannot meet the expectations of their parents. Therefore, those with individualistic cultural orientation might resist parenting practices that contradict their developmental needs [72], further exacerbating parent–adolescent conflict and intensifying the frustration of basic psychological needs, ultimately negatively impacting their well-being. The result suggests that the impact of parental control on adolescents’ development should be viewed dialectically based on the consistency or discrepancy in parent–adolescent cultural orientation.

This finding implies that the alignment or mismatch in cultural orientations between parents and adolescents significantly affects how parenting behaviors influence adolescents’ well-being. Specifically, the effects of parenting behaviors (like autonomy support or control) may be amplified or diminished depending on whether the parent and adolescent share similar cultural values or not. This nuanced understanding emphasizes the importance of considering cultural context in psychological research and interventions that focus on family dynamics and adolescent development.

### 4.3. Implications for Family Education Intervention Programs

The findings of this study shed light on the impact of traditionally controlling parenting behavior on adolescent well-being from the perspective of parent–adolescent cultural orientation alignment, offering insights for family education programs. In times of rapid societal transformation, many parents still hold collectivist cultural orientations. At the same time, their children may have shifted towards individualist orientations, leading to increased cultural orientation discrepancies and conflicts within families. Additionally, parents with collectivist orientations might continue traditional high-control parenting behavior and attribute any psychological or behavioral issues of individualistically oriented adolescents to character flaws or lack of resilience, exacerbating the frustration of psychological needs and decreasing well-being and potentially leading to higher rates of depression among these adolescents.

This necessitates schools or relevant authorities guiding parents to understand the differences in parent–child cultural orientations and their impact on adolescent well-being. It is essential to emphasize that cultural transformation is a general trend and that traditional high-control parenting can negatively affect the psychological development of adolescents, particularly when there is a mismatch in cultural orientations. Therefore, schools or related authorities should assist parents in adopting positive parenting concepts and adjusting their parenting behaviors appropriately to the times and their children’s cultural orientations. This approach can promote healthier psychological development and enhance the well-being of adolescents, contributing to more harmonious family dynamics and societal well-being.

### 4.4. Limitations and Future Directions

This study has several limitations: First, it was based on the level of parental involvement, ensuring quality interaction between either the father or mother and the child. The research examined the cultural orientation, parenting behavior, and outcomes of one parent. Future research could investigate both fathers’ and mothers’ cultural orientations and parenting behaviors to explore the impact of cultural orientation differences between parents within the family on parenting and adolescent development.

Second, this study adopted a cross-sectional design, which means the results cannot determine causality effects. Future research might conduct longitudinal studies, taking pre- and post-measurements of adolescent well-being, to more intricately examine the impact of similarities or differences in parent–adolescent cultural orientation on changes and fluctuations in adolescent well-being. This approach would allow for a more detailed understanding of the dynamics between cultural orientation, parenting behavior, and adolescent development over time, contributing to more effective and nuanced strategies in family education and intervention programs.

## 5. Conclusions

The conclusions of this study are as follows: (1) Adolescent-perceived parental autonomy support positively predicted the satisfaction of adolescents’ basic psychological needs, thereby enhancing adolescents’ well-being levels. Conversely, adolescent-perceived parental control significantly predicted the frustration of adolescents’ basic psychological needs, thereby reducing their well-being levels. (2) When both parents and adolescents share a collectivistic cultural orientation, high parental control significantly frustrated adolescents’ basic psychological needs, but it did not negatively affect their well-being. However, when parents are collectivists but adolescents are individualists, high parental control would significantly induce the frustration of basic psychological needs, thus further impairing adolescents’ well-being.

These findings highlight the importance of understanding the nuanced relationships between parenting behaviors, cultural orientation alignment, and adolescents’ well-being. Moreover, the present findings emphasize the need for parents and those involved in adolescent development to consider cultural contexts and the individual characteristics of adolescents when implementing or advising on parenting strategies. This is especially pertinent in diverse societies undergoing rapid cultural norms and values changes. This study contributes to the body of knowledge on parenting, cultural psychology, and adolescents’ development, offering insights that can inform future research, as well as practical applications in family education and policy making.

## Figures and Tables

**Figure 1 behavsci-14-00193-f001:**
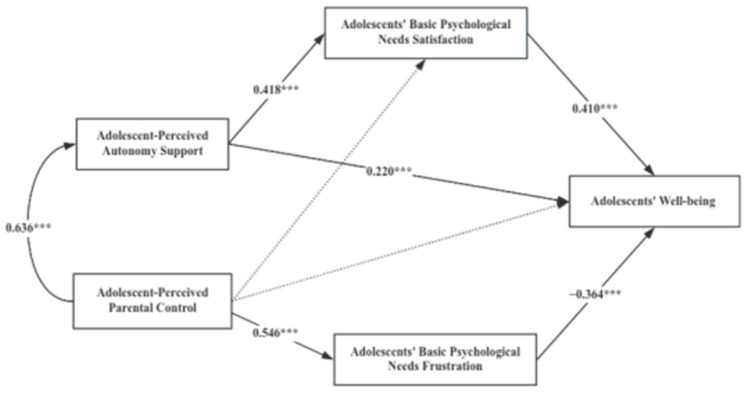
Diagram of the Mediating Role of Adolescent Basic Psychological Needs between Parenting Behaviors and Adolescent Well-Being; *** *p* < 0.001.

**Figure 2 behavsci-14-00193-f002:**
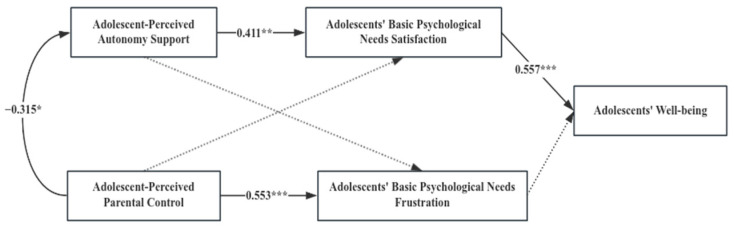
Path Diagram of Variables under Parental Collectivism and Adolescent Collectivism; * *p* < 0.05, ** *p* < 0.01 and *** *p* < 0.001.

**Figure 3 behavsci-14-00193-f003:**
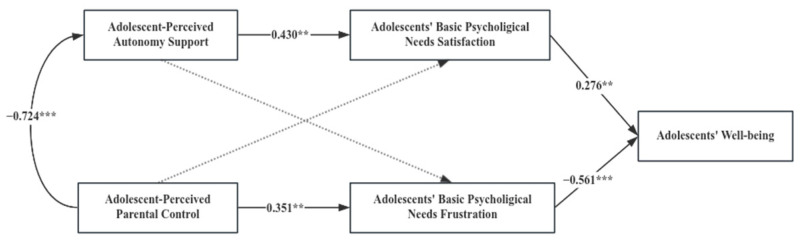
Path Diagram of Variables under Parental Collectivism and Adolescent Individualism; ** *p* < 0.01 and *** *p* < 0.001.

**Figure 4 behavsci-14-00193-f004:**
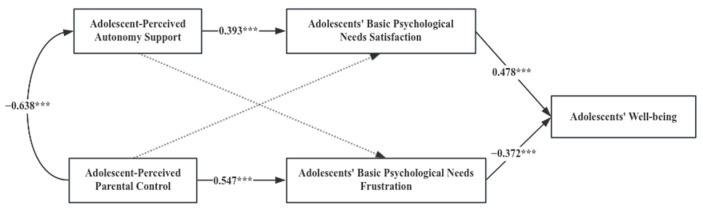
Path Diagram of Variables under Parental Individualism and Adolescent Individualism; *** *p* < 0.001.

**Figure 5 behavsci-14-00193-f005:**
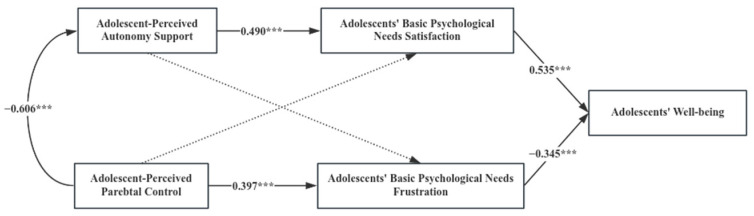
Path Diagram of Variables under Parental Individualism and Adolescent Collectivism; *** *p* < 0.001.

**Table 2 behavsci-14-00193-t002:** Indirect effects of basic psychological needs.

Pathways	Effect Size	SE	95% CI	
			Lower	Upper
APAS → ABPNS → AWB	0.171	0.029	0.117	0.232
APAS → ABPNF → AWB	−0.199	0.027	−0.256	−0.147

Note: APAS = adolescent-perceived autonomy support; APPC = adolescent-perceived parental control; ABPNS = adolescent basic psychological needs satisfaction; ABPNF = adolescent basic psychological needs frustration; AWB = adolescents’ well-being.

**Table 3 behavsci-14-00193-t003:** The multi-group analysis results.

Model	CMIN	DF	CFI	RMSEA	ΔCMIN	ΔDF	*P*
Unconstrained	89.724	36.000	0.958	0.048			
Structural weights	102.754	41.000	0.952	0.049	13.030	5	0.023
Structural residuals	111.948	47.000	0.949	0.046	22.224	11	0.023

**Table 4 behavsci-14-00193-t004:** Standardized path coefficients in different parent–adolescent cultural orientations.

Path	Ⅰ. P-C and A-C		Ⅱ. P-C and A-I		Ⅲ. P-I and A-I		Ⅳ. P-I and A-C	
	β	C.R.	β	C.R.	β	C.R.	β	C.R.
APAS → ABPNS	0.411	3.020 **	0.430	0.295 **	0.393	6.844 ***	0.490	5.072 ***
APPC → ABPNF	0.553	4.402 ***	0.351	0.314 **	0.547	10.243 ***	0.397	3.991 ***
APPC → ABPNS	0.085	0.623	0.036	0.027	−0.111	−1.926	−0.126	−1.306
APAS → ABPNF	0.231	1.836	−0.267	−0.220	−0.036	−0.676	−0.195	−1.957
ABPNS → AWB	0.557	4.948 ***	0.276	0.330 **	0.478	12.302 ***	0.535	7.775 ***
ABPNF → AWB	−0.212	−1.887	−0.561	−0.556 **	−0.372	−9.592 ***	−0.345	−5.014 ***

Note: ** *p* < 0.01, and *** *p* < 0.001. P-C = parental collectivism; P-I = parental individualism; A-C = adolescent collectivism; A-I = adolescent individualism; ABPNS = adolescents’ basic psychological needs satisfaction; ABPNF = adolescents’ basic psychological needs frustration; AWB = adolescents’ well-being.

**Table 5 behavsci-14-00193-t005:** Critical Ratios for Differences between Parameters.

	APAS → ABPNS	APPC → ABPNF	APPC → ABPNS	APAS → ABPNF	ABPNS → AWB	ABPNF → AWB
Ⅰ-Ⅱ	−0.354	−1.476	−0.308	−0.354	−1.135	−3.154
Ⅰ-Ⅲ	−0.262	−0.308	−1.239	−0.262	0.289	−1.734
Ⅰ-Ⅳ	0.150	−1.19	−1.219	0.150	0.857	−1.443
Ⅱ-Ⅲ	0.197	1.673	−0.951	0.197	1.736	2.421
Ⅱ-Ⅳ	0.599	0.448	−0.949	0.599	2.098	2.143
Ⅲ-Ⅳ	0.616	−1.371	−0.163	0.616	0.869	0.087

Note: Ⅰ = parents and adolescents are both collectivists; Ⅱ = parents are collectivists but adolescents are individualists; Ⅲ = parents and adolescents are both individualists; Ⅳ = parents are individualists but adolescents are collectivists.

## Data Availability

The data presented in this study are available upon request from the corresponding author.

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
