# Peer review of "The Relationship between Parenting Behaviors and Adolescent Well-Being Varies with the Consistency of Parent–Adolescent Cultural Orientation"

_behavsci, 2024, doi:10.3390/bs14030193_

Round 1
Reviewer 1 Report
Comments and Suggestions for Authors
I was pleased to read the manuscript entitled "The relation between Parenting behaviors and Adolescent Well-being Vary as the Consistency of Parent-adolescent Cultural Orientation" and to review it.
The current study estimates the mediating role of basic psychological needs satisfaction between parental autonomy support and adolescents’ well-being, as well as the mediating role of basic psychological needs frustration between parental control and adolescents’ well-being. A random sample of Chinese adolescents and their parents dyads (N=644) was interviewed and conventional methods of analysis, in general, based on structural equation modeling, were used. The content of the article is suitable for Special Issue: "Parenting and Positive Development for Children and Adolescents".
The article is written in a typical format as follows.
Title – the title is informative and it more-or-less reflect the manuscript. However, its meaning can be understood only after familiarizing with the content of the article. So maybe it should be improved to have a more scientific tone and avoid vague terms (for example, "Cultural Orientation"). A possible title alternatives could be: "The mediating role of basic psychological needs in the relation between parenting behaviors and adolescent well-being", or "Consistency of parent-adolescent orientation towards young generation autonomy and its impact on adolescent well-being".
Abstract – the abstract is more or less complete and adequately reflects the content of the manuscript. However, the aim of the research must be clearly formulated. I recommend breaking the first sentence of the abstract into several sentences, defining the problem and the aim of the research. The last sentence of the abstract asks to answer what the effect is aimed at (... on adolescent well-being?).
Keywords – the list of keywords must be more specific. at least it should include "Adolescents".
Introduction – the Introduction provide sufficient theoretical background for the study. The rationale of the study is well described and the study problem is stated clearly. The introduction is structured logically and the text is fluent. Relevant and unbiased literature was used. My comments:
lines 137--130: The second hypothesis is given, so what was the first hypothesis?
lines 130-133: The text here must be reviewed from a linguistic point of view.
Materials and Methods – the study utilised a quantitative approach, the methods are quite clearly described and are appropriate to answer the proposed research questions. However, I suggest:
1. The authors appear to be using original questionnaires that they collected from a number of other studies (subsections 2.2.1 and 2.2.2 should cite sources) and adapted them for their study. It is necessary that these questionnaires in English are submitted in additional files.
2. Information about the internal consistency coefficient of the questionnaire can be removed, because it is not used further.
Results – in general, results are clearly organized and presented.
Discussion – the structure of the Discussion is clear. The interpretations is appropriate and is supported by the results. The study findings are discussed with relevant literature and within the limits of the study.
Conclusions – are supported by the results.
References – References must be submitted according to the journal requirements. A lot of corrections are needed here.
Summarizing this review, it can be said that the authors of the article have a professional insight into the problem under investigation, so it was difficult for me to find anything wrong with this article. The comments presented are more technical in nature than substantive.
Thank you for considering my opinion. I encourage authors to keep on working to improve the manuscript.
Comments on the Quality of English LanguageModerate editing of English language required
Author Response
Dear Reviewer,
Thank you very much for taking the time to review this manuscript entitled "The relation between Parenting behaviors and Adolescent Well-being Vary as the Consistency of Parent-adolescent Cultural Orientation."
In order to explain the limitations of parenting behaviors in the period of Chinese cultural transformation, this study examined the mediating role of adolescent basic psychological needs satisfaction and frustration between parental rearing behavior and adolescent well-being. Moreover, the moderating role of the consistency in parent-adolescent cultural orientation in the impact of parenting behaviors on adolescent well-being was also studied. Six hundred forty-four parent-adolescent dyads completed self-report surveys. Consistent with our hypothesis, adolescents' basic psychological needs satisfaction and frustration play a mediating role between parenting behaviors and adolescents ' well-being. At the same time, the consistency of parent-adolescent cultural orientation also moderates the impact of parenting behavior on adolescent well-being.
I will reply to your suggestions one by one from the following aspects.
Title--Title 1 ' The mediating role of basic psychological needs in the relationship between parenting behaviors and adolescent well-being ' only covers research hypothesis 1 and does not highlight the focus and innovation of the article; title 2 ' Consistency of parent-adolescent orientation towards young generation autonomy and its impact on adolescent well-being. The cultural orientation between parents and children includes not only autonomy, but also individualism values such as competition and privacy, as well as collectivism values such as relevance and filial piety. Although this title contains part of the article, it does not reflect a more comprehensive culture.
In addition, Although the title of this article contains the seemingly vague term ' cultural orientation ', it does find that it is positioned in the individualism-collectivism cultural orientation when searching for this term. At the same time, the title is determined by our comprehensive consideration of the focus, innovation and research conclusions of this article. Therefore, we prefer to use existing title.
Abstract –When writing the first draft, we tried to split it into several independent sentences, but through comparison, we found that when the research purpose and the research question are placed in the same sentence, the logical relationship between the two is stronger. Therefore, we are more inclined to present the research purpose and the question in the same sentence.
Through the content of the article, we can know that the effects of parenting behavior specifically refers to the impact of parenting behaviors on adolescent well-being based on the SDT framework. Therefore, in the last sentence of the abstract, although we do not directly express the impact of parent-child cultural differences on adolescent well-being, we indirectly express this meaning through the role of parenting behavior.
Keywords --The keyword part plus the main group ' Adolescent ' will indeed be clearer. Therefore, we add adolescent before the basic psychological needs and well-being, so as to clarify the survey group of variables.
Introduction--In the process of putting forward the research hypothesis, the introduction part focused on the role of parent-adolescent cultural consistency in the relationship between parenting behavior and adolescent well-being, but ignored the hypothesis that basic psychological needs satisfaction and frustration plays a mediating role between parenting behavior and adolescent well-being. Thanks to you, we discover the problem in time and modify it accordingly, adding the inference process of hypothesis 1 to the ' The present study' section.
Materials and Methods –In the part of materials and methods, because the existing literature on cultural orientation measurement tools is too complicated and cumbersome, the tools used in the adaptation of parents ' cultural orientation questionnaire in 2.2.1 are not listed in detail. Moreover, the cultural orientation questionnaire for adolescents in 2.2.2 is adapted from the cultural orientation questionnaire for parents in 2.2.1, and the tools used are the same as those in 2.2.1, so it is also skimmed. Based on your suggestion, we submit the questionnaire in the form of additional documents so that you and the editor-in-chief can have a further understanding of the questionnaire measuring the cultural orientation of parents and adolescents.
Secondly, in order to verify the reliability of the relevant measurement tools in the survey group, we have presented the internal consistency coefficient in the measurement tool part. Although it is not further used later, it can also be used as the reliability evidence of the measurement tool in the group.
References –For the format of the reference, we have referred to the journal requirements for correction. Thank you for your correction.
Comments on the Quality of English Language-- For the use of English language in this paper, we have modified it through two native English speakers after the first draft is written.
Thank you again for your valuable comments on this article! The further improvement of this article is inseparable from your help. If there are other suggestions, we are willing to communicate with you.
Best wishes!
Tixiang Yang

Reviewer 2 Report
Comments and Suggestions for Authors
Dear Editor,
Thank you for letting me review this interesting and important paper: The relation between Parenting behaviours and Adolescent Well-being Vary as the Consistency of Parent-adolescent Cultural Orientation.
My suggestions:
· Make some more subtitles in the introduction and results.
· What was included in the data cleaning?
· Add to the discussion that the COVID-19 pandemic might have impacted their answers.
· Is it important if the parents pay for the education?
· Is it black or white like you describe differences in cultures?
·
Best wishes
Author Response
Dear Reviewer,
Thank you very much for taking the time to review this manuscript entitled "The relation between Parenting behaviors and Adolescent Well-being Vary as the Consistency of Parent-adolescent Cultural Orientation."
In order to explain the limitations of parenting behaviors in the period of Chinese cultural transformation, this study examined the mediating role of adolescent basic psychological needs satisfaction and frustration between parental rearing behavior and adolescent well-being. Moreover, the moderating role of the consistency in parent-adolescent cultural orientation in the impact of parenting behaviors on adolescent well-being was also studied. Six hundred forty-four parent-adolescent dyads completed self-report surveys. Consistent with our hypothesis, adolescents' basic psychological needs satisfaction and frustration play a mediating role between parenting behaviors and adolescents ' well-being. At the same time, the consistency of parent-adolescent cultural orientation also moderates the impact of parenting behavior on adolescent well-being.
I will reply to your suggestions one by one from the following aspects.
The first suggestions of making some more subtitles in the introduction and results--First of all, since the introduction part has only three paragraphs and the first sentence of each paragraph is equivalent to the main idea and sub-title of the paragraph, the sub-title and the paragraph meaning of each paragraph are repeated in the introduction part. In addition, the results section also has corresponding subtitles to clearly clarify the relationship between variables. In summary, in order to make the structure of the article smooth and clear. In the introduction and conclusion part, we still retain the original structural layout.
The part of the data cleaning--We cleaned the data according to the following criteria:
1. Delete the questionnaire whose informed consent right is ' No '.
2. Delete the missing answer, wrong answer questionnaire.
- Remove the questionnaire with a total score of 4 points or more of lie detection questions in the Parental Cultural Orientation Questionnaire
- Questionnaires with a total lie detection score of 4 or above were deleted from the questionnaires related to adolescents ' cultural orientation, perceived parenting behavior, basic psychological needs satisfaction and well-being.
5 Whether the student ID number or student number and the class filled in the Parental Cultural Orientation Questionnaire and the youth version of the cultural orientation questionnaire can be one-to-one correspondence. If not, parents and the youth version of the cultural orientation questionnaire will be deleted.
This part has been added to the ' Data analysis' section.
The third suggestion of adding to the discussion that the COVID-19 pandemic might have impacted their answers--The organizational ability and efficiency of Chinese collectivism in the face of disasters have played an important role in the process of overcoming the epidemic. The high degree of self-discipline, sense of responsibility and protection of family and their own lives of most people also play a key role in the epidemic. This is a point that we did not take into account when adapting the questionnaire, and we believe that it is more accurate as a future research direction than as a discussion part. It is hoped that future research can measure the cultural orientation of parents and adolescents through their views on the promulgation of national policies and their attitudes towards the epidemic, so as to examine the cultural orientation of parents and adolescents from a broader perspective.
The importance of parents paying for the education--Parents ' economic resources may affect their parenting concepts and behaviors, which in turn will affect their children's development. Studies have confirmed that the economic status of parents will affect their parenting concepts and parenting behaviors. Parents with higher socioeconomic status will pay attention to the development of individualism such as children 's autonomy and independence, while parents with lower socioeconomic status will pay attention to the cultivation of collectivism such as filial piety and obedience. Therefore, whether parents have the ability to pay for education is important for their children 's development.
About the question, Truth be told, we failed to understand your question and hope to have the opportunity to communicate with you further.
The last question: Is it black or white like you describe differences in cultures?
On this issue, we can start with the literature related to race and parenting concepts. In the existing literature on race and parenting concepts, we can find that there are racial differences in parenting concepts. Under the premise of controlling economic status, parents of white groups mainly adopt an independent cultural model in the process of parenting, that is, parents will regard their children as separate individuals and support their children 's self-development and self-realization. In the process of parenting, parents of black groups will be more dominated by interdependent cultural models, regard their children as part of themselves, and emphasize their children 's relationship with others. Therefore, at the macro level, the cultural differences between the white group and the black group are similar to those in the text. However, due to the personal cultural value orientation, it may also be closely related to the macro-cultural background and the level of regional development. Based on this, we can make such a speculation that after controlling the economic level, the cultural differences of the black population may be closer to this article ; the white group may be opposite to the cultural differences in this paper due to the cultural background of individualism.
Thank you again for your valuable comments on this article! The further improvement of this article is inseparable from your help. If there are other suggestions, we are willing to communicate with you.
Best wishes!
Tixiang Yang
